# The ChatGPT Effect: Investigating Shifting Discourse Patterns, Sentiment, and Benefit–Challenge Framing in AI Mental Health Support

**DOI:** 10.3390/bs15091172

**Published:** 2025-08-28

**Authors:** Sanguk Lee, Minjin (MJ) Rheu, Jie Zhuang

**Affiliations:** 1Department of Communication Studies, Texas Christian University, Fort Worth, TX 76109, USA; 2Division of Media & Communication, Hankuk University of Foreign Studies, Seoul 02450, Republic of Korea; 3School of Communication, Loyola University Chicago, Chicago, IL 60660, USA

**Keywords:** large language models, mental health, AI for mental health, Reddit

## Abstract

AI has the potential to enhance mental health by scaling support. However, its implementation brings uncertainties and challenges that require careful review to ensure safety. This study examined evolving public views on AI mental health support by analyzing relevant Reddit posts (*n* = 517). Following the release of ChatGPT in 2022, discussions about AI in the context of mental health surged, with a noticeable shift in preference toward large language models (LLMs) over conventional therapy chatbots. Users appreciated AI for its emotional support, companionship, and accessibility, while also expressing concerns about adverse effects and lack of conversational depth and emotional connection. Distinct patterns in how benefits and challenges were discussed emerged between experienced and non-experienced AI users, as well as between AI-focused and mental health-focused communities. AI-experienced users acknowledged both the benefits and limitations, whereas AI communities emphasized the positives and mental health communities highlighted the lack of conversational depth. These findings underscore the need for tailored communication strategies to set realistic expectations about the utility of AI in mental healthcare among different stakeholders. This research provides insights into developing ethical AI systems that complement traditional care while addressing current limitations.

## 1. Introduction

Mental health—referring to emotional and psychological well-being—is a fundamental aspect of human well-being, influencing individuals’ ability to navigate daily life, engage in social interactions, and make meaningful contributions to society. Despite its critical importance, mental health challenges remain widespread. Current data indicate that one in eight individuals globally lives with a mental disorder, while in the United States, more than one in five adults experiences mental illness ([29]; [37]). The prevalence of mental health conditions, coupled with a lack of resources and infrastructure in support systems, results in significant gaps in mental health care accessibility ([13]; [15]; [22]).

Various interventions have been developed to address mental health challenges, such as peer support programs and community-based health centers ([33]; [35]). Although these approaches help mitigate some demands of mental health support ([35]), their heavy reliance on human resources still limits their expansion and scalability ([9]).

Recent advancements in artificial intelligence (AI), specifically large language models (LLMs), which are capable of generating natural human language, offer promising avenues to address these scalability and accessibility issues. In this study, AI broadly refers to language-based tools and systems that interact with humans through natural language, ranging from simple linguistic algorithms to highly sophisticated LLMs. Advanced AI such as LLMs, in particular, demonstrate capabilities such as emotion recognition ([39]), empathetic response generation ([2]), and provision of personalized, accessible, non-judgmental support interventions ([23]; [31]).

Despite these promising capabilities, AI-driven mental health support poses challenges including limitations in genuinely understanding human emotions ([27]), algorithmic biases, privacy concerns, and risks of over-dependence ([17]). Furthermore, there is a lack of scientific evidence regarding whether using LLMs to address mental health challenges is safe and reliable. Reflecting these concerns, the American Psychological Association (APA) recently issued a cautionary statement during a Federal Trade Commission (FTC) panel discussion regarding the use of such technology ([3]). These contrasting perspectives highlight the need to critically investigate user perceptions and interactions with AI-driven mental health tools.

To explore the potential benefits and pitfalls of this emerging trend, this study analyzes unstructured text data scraped from Reddit, a widely used social media platform where individuals voluntarily share their candid perspectives on AI-based mental health support. Social media provides unique insights into public sentiment and evolving perspectives that might be difficult to capture through traditional research methodologies. In particular, the introduction of ChatGPT marked a significant shift in public discourse regarding AI in mental health support. By analyzing time-stamped Reddit threads, this study examines shifts in public perception before and after ChatGPT’s deployment, identifying key opportunities and challenges associated with AI mental health interventions.

This research contributes to the growing literature on AI and mental health by providing empirical insights into user concerns, expectations, and experiences. Understanding these perspectives can provide valuable insights for developing ethical, effective, and human-centered AI systems that complement traditional mental health care.

### 1.1. The Transformation of AI in Mental Health Support: Before and After LLMs

AI integration into mental health support began in the 1960s with ELIZA, an early language system developed by Joseph Weizenbaum at MIT. Modeled after a Rogerian psychotherapist, ELIZA used pattern matching to reformulate user statements as questions, creating an illusion of understanding despite lacking true comprehension ([36]). While limited, ELIZA demonstrated the potential for computer-mediated dialogue to facilitate self-reflection. As technology advanced, AI-driven mental health applications expanded significantly with mobile technologies in the 2010s. Chatbots like Woebot and Wysa used rule-based responses to deliver cognitive-behavioral therapy interventions, providing accessible, stigma-free support for anxiety and depression ([1]). Despite these innovations, early AI systems remained constrained by their inability to provide the natural, flexible, and contextually adaptive interactions essential for meaningful mental health support.

The recent emergence of large language models (LLMs) has the potential to fundamentally transform the delivery of mental health support. Unlike their predecessors, which operated based on restricted language rules, LLMs leverage extensive datasets and advanced deep neural networks to generate more natural, contextually relevant, and human-like responses. This technological breakthrough has shown promising implications for AI’s role in mental health care. Studies have indicated that LLMs can accurately recognize human emotions embedded in texts ([39]), generate empathetic messages that are accurate, clear, and relevant in response to emotionally challenging queries ([24]), and even surpass the quality of human-generated empathetic messages ([2]; [19]).

Despite these capabilities, public and professional reactions to AI-driven mental health support remain mixed. A recent survey suggests while individuals from mental health communities expressed their optimism about AI support systems enhancing personalized care, accessibility, and efficiency, they also expressed concerns about the lack of human connection, the risk of misdiagnosis, and privacy issues ([7]). Similarly, mental health professionals expressed both enthusiasm for AI’s potential and apprehension regarding risks and misuses, ethical challenges, and data security ([7]).

LLMs are making a significant impact across industries and systems, including mental health care. Among LLMs, ChatGPT stands out as a particularly notable milestone, representing one of the first widely accessible commercial LLMs to enter public consciousness. The release of ChatGPT in November 2022 may serve as a critical temporal marker for understanding shifts in public perception and engagement with AI-based mental health support. To capture this evolving discourse, our study examines trends in social media posts, analyzing changes in frequency, the types of AI applications discussed in mental health contexts, and sentiment toward AI before and after ChatGPT’s release.

RQ1: How have (a) the prevalence of AI discussions, (b) the types of AI applications, and (c) sentiments toward AI in mental health changed before and after the release of ChatGPT?

### 1.2. Potential Benefits and Limitations of Using AI in Mental Health Care

As mentioned earlier, the introduction of AI in mental healthcare presents both significant opportunities and challenges. Table 1 lists the benefits and limitations along with their definitions. Most items (*n* = 10) were identified through a narrative review of literature discussing benefits and challenges of AI use in mental health, while others were derived from our own data analysis (*n* = 4; detailed descriptions are available in the methods section). Below, we provide narrative descriptions of how each category benefits or poses risks for individuals who use AI for mental health care.

By offering greater *accessibility and availability*, AI-driven interventions can substantially benefit individuals lacking access to traditional support systems, delivering assistance anytime, anywhere, and at lower cost ([23]; [26]). With AI’s capability for social interaction and contextual learning, it can provide *advice tailored to individual needs*, assist *self-reflection*, while offering *emotional support* and *companionship* ([12]; [23]). The *non-judgmental nature* of AI is another reason that attracts individuals seeking mental health support. Many people hesitate to seek help due to fears of stigma and judgmental reactions to their mental health concerns ([6]). AI can reduce such fears, as people tend to perceive it as inherently more objective and non-judgmental ([34]), encouraging earlier help-seeking behavior ([20]; [31]).

However, AI also has several critical limitations in mental health support. Despite sharing the same content, support messages from AI can be perceived as less effective than those from humans ([21]; [25]). This perception can stem from machine-heuristics—where people view machines as cold and impersonal ([38])—making AI *insufficient for deep conversations and building emotional connections*. The potential for *adverse effects* presents another significant concern. While AI companies strive to enhance safety, current systems still generate biased, hallucinated, or harmful content that might exacerbate mental health conditions ([5]; [17]). *Over-reliance* on AI poses an additional risk—its constant availability, while beneficial, can lead to excessive dependence. AI’s tendency to be compliant and tell users what they want to hear may discourage them from seeking essential human connections ([12]; [23]) and may instill unrealistic expectations from other relationships. Furthermore, *technical and privacy issues* create substantial implementation challenges. Users share private and sensitive information with the chatbot, and the handling of such data raises serious privacy concerns. Moreover, technical instability such as abrupt system downtime or connectivity issues could disrupt critical moments of support sessions ([17]; [20]).

While previous research has documented these various advantages and limitations of AI in mental health applications, empirical evidence regarding how users actually discuss these factors remains limited. This gap hinders the identification of genuine user concerns about AI-based mental health interventions and may limit the development of human-centered AI. Social media platforms provide valuable access to spontaneous, authentic discussions that reveal which aspects of AI mental health support users most frequently discuss. By analyzing social media data based on a comprehensive list of benefits and challenges of AI use in mental health, this research seeks to identify patterns in these discussions.

RQ2: What are the benefits and challenges of using AI in mental health that are frequently discussed among the users?

Beyond understanding general patterns in social media discourse, we are particularly interested in how perceptions differ between those with direct experience using AI for mental health support and those without such experience. Experienced users might discuss benefits and challenges based on firsthand interactions, whereas non-users may rely on generalized beliefs or secondhand information, potentially leading to divergent perspectives about AI-based mental health support ([20]). This comparison forms the basis of our third research question.

RQ3: How do patterns of discussing benefits and challenges vary between users who directly interact with mental health-related AI and non-users?

Furthermore, different online communities may exhibit unique patterns of discussion regarding the use of AI for mental health support. A previous study reveals that general attitudes toward AI are associated with individuals’ levels of AI knowledge, AI learning anxiety, and computer use ([14]). Given that members of AI and mental health communities may differ in their knowledge and interests regarding AI technologies, their attitudes and discourse patterns toward AI diverge. For example, AI communities may emphasize the positive aspects of AI technologies, reflecting their generally favorable views, while mental health communities may focus more on the limitations of AI in meeting their emotional and therapeutic support needs. Examining variations between dedicated AI and mental health communities offers insight into how technical enthusiasm, therapeutic priorities, and practical needs influence users’ engagement with—and judgment of—AI interventions. Our final research question addresses this dimension.

RQ4: How do patterns of discussing benefits and challenges vary between AI communities and mental health communities?

## 2. Methods

### 2.1. Data Collection and Filter

IRB review was exempted for collecting publicly accessible Reddit posts in which all content is viewable without registration or login. No interaction with users occurred, and no private or direct messages were accessed. All data are stored on a local computer protected by password authentication, with usernames removed before data analysis. We did not report any individual posts in our findings to prevent potential identification of users. All results are presented as aggregated data only, ensuring individual voices cannot be traced back to specific users.

We collected data on 18 October 2024, using RedditHarbor, an open-source data collection tool ([30]). In this study, AI refers to language-based AI tools and systems that interact with humans through natural language, and mental health broadly refers to emotional and psychological well-being. Within the scope, we targeted collecting posts that discuss AI in a context related to mental health, including specific use cases for mental health care but also general discussion of how AI use affects their emotional and psychological health.

Our sampling strategy involved a systematic selection of relevant subreddit communities. First, we identified subreddits focused on either “mental health” or “AI” discussions using these two keywords. To ensure sufficient engagement, we only included communities with over 1000 members at the time of data collection. We excluded mental health communities specific to particular countries, cultures, or demographic groups, which could introduce potential sampling bias. Moreover, since our research focuses specifically on language-based AI services, we excluded AI communities primarily discussing non-language AI applications (e.g., image or video generation). Using these criteria, we identified 32 subreddit communities: eighteen mental health-focused communities (e.g., r/mentalhealth) and fourteen AI-focused communities (e.g., r/ChatGPT); we categorized these as “mental health communities” and “AI communities,” respectively. Appendix A provides the list of the included communities and their approximate membership sizes.

Our data collection strategy employed distinct keyword sets tailored to each community type in order to collect potentially relevant posts concerning both AI and mental health. For the 18 mental health communities, we searched using AI-related keywords (e.g., ChatGPT, chatbot), while for the 14 AI communities, we used mental health-related keywords (e.g., mental health, anxiety). Appendix A presents the list of keywords. From these searched posts, we collected information including unique post IDs, author IDs, titles, text content, creation dates, community names, and engagement metrics (comment counts and upvotes).

Our data collection yielded 2097 posts. We removed 202 posts that contained no text. Then, we employed a fine-tuned GPT-4o model as an automated content analyzer to further filter out posts irrelevant to the intersection of AI and mental health. This analysis identified 599 posts that discussed AI in relation to mental health or emotional well-being. We further excluded 78 posts that either purely promoted AI mental health applications, simply shared prompts for creating AI mental health personas without commentary, or merely displayed AI outputs without user opinions as well as four irrelevant posts. The final valid 517 posts were included for the main analysis. These target posts span from 24 October 2014 to 18 October 2024. The detailed information about the content analysis and the validity test is presented in the next section.

### 2.2. Automatic Content Analysis

We utilized a fine-tuned version of OpenAI’s GPT-4o model to identify valid posts and extract multiple semantic features. Fine-tuned LLMs offer advantages over generic pre-trained models (e.g., GPT-4o) for tasks like content analysis, as they are further trained on domain-specific data to adjust model weights and enhance task-specific performance. Creating a fine-tuned model requires training data that adequately represent the variables of interest. To generate such data, we first used a generic GPT-4o model to identify posts potentially relevant to our study (i.e., discussing AI in mental health contexts). From these, we sampled a subset of Reddit posts (*n* = 70) for manual coding. Three coders independently coded an initial subset of these posts. Discrepancies were discussed, and the coding scheme was refined until consensus was achieved. The remaining posts were then jointly coded using the finalized scheme in Table 2, with disagreements resolved through discussion.

The human-annotated data served as ground truth to fine-tune GPT-4o. To assess model performance, we trained a GPT-4o model on a subset of the ground truth data (*n* = 50), reserving the remaining samples (*n* = 20) as a true holdout test set. On this holdout, the model achieved an accuracy of 0.87, with precision, recall, and F1-scores of 0.71, 0.71, and 0.70, suggesting the model could generalize to unseen data (See Appendix A for details). To make full use of the available annotations, we then fine-tuned GPT-4o on the complete ground truth dataset (*n* = 70) and applied this model to code the full dataset. This model achieved a training loss of 0.001 and validation loss of 0.032. While interpretation of these values is task-dependent, the relatively low and similar losses may suggest that the model fit the data effectively without obvious signs of overfitting.

The coded variables include *relevance* indicating whether posts discussing AI in relation to mental health, *promotion and prompt* to filter out purely promotional posts or prompts, *AI applications* indicating which type of AI applications people used, *user interaction* indicating whether a user directly interacts with AI related to mental health, and *dominant sentiment toward AI* indicating dominant sentiment expression toward AI. The coding scheme, which contains more detailed information such as coding instructions, variable definitions, and class criteria, is available in Appendix A.

Furthermore, we coded both the benefits and challenges of using AI in the context of mental health. To create a comprehensive list, we employed both top-down and bottom-up approaches. In the top-down approach, we reviewed prior studies to identify potential benefits and challenges associated with AI in mental health care. In the bottom-up approach, we explored the data with the assistance of a pre-trained GPT-4o model. Specifically, we provided the model with a subset of posts related to AI and mental health identified during the initial data exploration stage and instructed it to extract mentions of benefits and challenges. Researchers then reviewed the model’s output and identified additional themes that were not documented in previous studies but appeared frequently in our data. All benefit categories were captured through the top-down approach, whereas two challenge categories—namely, lack of human-like interaction and lack of professional qualifications—emerged from the data-driven approach and were added to our coding list.

Therefore, the final list of benefits includes accessibility and availability, assistance with self-reflection, companionship, emotional support, non-judgmental attitude, personalized advice and support, and other benefits. The list of challenges comprises adverse effects, insufficient depth and emotional connection, lack of human-like interaction, lack of professional qualifications, overreliance, technical and privacy issues, and other challenges. These variables were dichotomously coded as “yes” or “no,” depending on the presence of each component in a post. It is noteworthy that a single post may mention multiple benefits and challenges.

### 2.3. Data Analysis

We conducted descriptive analysis to investigate the trend of discussing AI in the context of mental health (RQ1) and the frequency of benefits and challenges of such AI use (RQ2). To examine the differences in how benefits and challenges were discussed across user groups, we conducted two sets of logistic regression analyses. For our third research question (RQ3), we analyzed how AI experience influenced the mention of each benefit and challenge by using logistic regression models where the independent variable was user experience (AI use vs. no AI use, with no AI use as the reference category) and the dependent variables were the presence or absence of each specific benefit and challenge. Additionally, to address our fourth research question (RQ4), we performed similar logistic regression analyses to compare discussion patterns between different community types, using community type as the independent variable (AI-focused vs. mental health-focused, with AI-focused community as the reference group) and the presence of each benefit or challenge as dependent variables. This analytical approach allowed us to systematically identify significant differences in how benefits and challenges were mentioned based on both user experience and community membership.

## 3. Results

### 3.1. AI and Mental Health Discussion Before and After ChatGPT

We investigated the trend of social media posts about AI and mental health over a 10-year period from October 2014 to October 2024. As illustrated in Figure 1, users’ interest in the intersection of AI and mental health increased dramatically following the release of ChatGPT on 30 November 2022. In our dataset, 74% of posts (*n* = 382) were collected during the post-ChatGPT era, compared to 26% (*n* = 135) from the pre-ChatGPT era. Although discussions peaked and then experienced a slight decline, the overall upward trend has persisted over the years. This pattern was evident not only among AI-focused communities but also within mental health communities. These findings indicate that public interest and conversation about the use of AI in mental health on social media have shifted significantly following the introduction of ChatGPT.

The distribution of AI applications mentioned in posts shifted significantly between the pre- and post-ChatGPT eras. Figure 2 illustrates the distribution of AI applications before and after ChatGPT release. The proportions for the “others” (e.g., reddit bots, snapchat bots, general AI bots) and “therapy chatbots” categories declined from 53% to 33% and from 24% to 13%, respectively. Despite the decline, “others” still occupies a substantial proportion. This might be attributed to the fact that posts, especially from mental health communities, did not specify the AI applications that users interacted with or encountered.

In contrast to others and therapy chatbots, mentions of “LLM” applications (e.g., GPT, Claude, Gemini, Llama) emerged at 25% following ChatGPT’s release, reflecting a growing interest in leveraging LLMs for mental health applications. Notably, the proportion of posts referencing “character-based chatbot” applications (e.g., Replika and Character.ai) increased from 23% to 29%. This increase suggests that these platforms, possibly incorporating LLMs into their services, continue to play a significant role in influencing users’ mental health well-beings.

There have been notable shifts in sentiment expressed toward AI in the context of mental health. Figure 3 illustrates the distribution of sentiment toward AI before and after ChatGPT release. In the post-ChatGPT era, people more frequently express sentiments about AI compared to the pre-ChatGPT period. This shift is reflected in the declining proportion of neutral sentiment from 42% to 33%, while positive sentiment increased from 24% to 29%, negative sentiment rose from 27% to 30%, and mixed sentiment grew from 7% to 9%.

### 3.2. Benefits and Challenges of AI Use in Mental Health

We examined the benefits and challenges that users expressed regarding AI applications in mental health. From this analysis to the end, we focus exclusively on posts from the post-ChatGPT era to capture opinions on more recent AI applications. Overall, users mentioned benefits (the number of benefit instances = 315) more frequently than challenges (the number of challenge instances = 199). Among the benefits, users most frequently appreciated AI’s ability to provide emotional support (25.7%), followed by companionship (20.6%). Accessibility and availability were also cited as key benefits (19.0%), while additional advantages included AI’s capacity to assist in self-reflection (13.7%), offer personalized advice (11.1%), and deliver non-judgmental support (9.2%). On the challenge side, adverse effects were by far the most mentioned challenge (46.2%). In addition, users mentioned AI’s insufficient depth and emotional connection as the second most challenging (20.6%). Other challenges included users’ overreliance on AI (11.6%), AI’s ability to display human-like interaction (9%), technical and privacy issues (7%), and a lack of professional qualifications (4.5%). Figure 4 illustrates distribution of benefits and challenges of AI use regarding mental health.

### 3.3. Variations in Benefit and Challenge Discourse: AI Users vs. Non-Users

We examined whether users’ perspectives differ between users who use AI in the mental health context and non-users. There were more posts in which authors explicitly indicated that they have used AI (*n* = 249) than posts where authors explicitly noted that they have not used AI (*n* = 38) in the mental health context. When comparing posts from these groups, posts from users with AI experience had approximately eleven times higher odds of mentioning companionship (*b* = 2.40, *SE* = 1.03, odds ratio (OR) = 10.98, *p* = 0.02) and nearly three times higher odds of mentioning emotional support compared to non-users (*b* = 1.18, *SE* = 0.55, OR = 3.26, *p* = 0.03).

Regarding challenges, although statistically insignificant, there were trends suggesting that posts from AI users mentioned adverse effects (*b* = 0.95, *SE* = 0.50, OR = 2.58, *p* = 0.06) and insufficient depth and emotional connection (*b* = 1.87, *SE* = 1.03, OR = 6.46, *p* = 0.07) more likely than posts from non-users. Due to quasi-complete separation—a condition in which the outcome variable shows no variation within a group—statistical results for some variables in the challenge and benefit dimensions were unstable and therefore not reported. Table 3 summarizes how AI interaction experience was associated with the discussion of benefits and challenges. Appendix A visually illustrate distributions of benefits and challenges between the two groups, respectively.

### 3.4. Variations in Benefit and Challenge Discourse: AI Communities vs. Mental Health Communities

We further investigated whether benefit and challenge discourse varied between posts written by users from AI communities and mental health communities. There were slightly more posts from AI communities (*n* = 196) than posts from mental health communities (*n* = 186). Posts from mental health communities were less likely to mention nearly all dimensions of positive benefits compared to posts from AI communities. Specifically, posts from mental health communities had only 32% of the odds of mentioning AI’s assistance with self-reflection compared to posts from AI communities (*b* = −1.13, *SE* = 0.37, OR = 0.32, *p* = 0.002). In other words, posts from AI communities had approximately 3.13 times higher odds (1/0.32 = 3.13) of mentioning it than posts from mental health communities.

Posts from mental health communities also had only 47% of the odds of mentioning accessibility and availability (*b* = −0.76, *SE* = 0.30, OR = 0.47, *p* = 0.01), 31% of the odds of companionship (*b* = −1.17, *SE* = 0.30, OR = 0.31, *p* < 0.001), 39% of the odds of emotional support (*b* = −0.95, *SE* = 0.27, OR = 0.39, *p* < 0.001), 37% of the odds of non-judgmental support (*b* = −0.98, *SE* = 0.43, OR = 0.37, *p* = 0.02), and 33% of the odds of personalized advice (*b* = −1.10, *SE* = 0.43, OR = 0.33, *p* = 0.006).

In contrast, the pattern of discussing challenges did not differ substantially between the two groups, except in the case of insufficient depth and emotional connection. Posts from mental health communities had nearly two and a half times the odds of mentioning this theme compared to posts from AI communities (*b* = 0.91, *SE* = 0.35, OR = 2.49, *p* = 0.009). Although not statistically significant, posts from AI communities tended to include more discussion of adverse effects than those from mental health communities (*b* = −0.45, *SE* = 0.24, OR = 0.54, *p* = 0.06). Due to quasi-complete separation where a predictor almost perfectly predicts the outcome variable, statistical results for some variables in the benefit and challenge dimensions were unstable and therefore not reported. Table 4 summarizes how community type was associated with the discussion of benefits and challenges. Appendix A visually illustrate distributions of benefits and challenges between the two groups, respectively.

## 4. Discussion

AI systems have the potential to enhance mental well-being by providing scalable support that complements human support. However, such implementation poses significant uncertainties and potential risks, requiring careful scrutiny for safe and reliable use. This research explored evolving public opinion on AI-driven mental health support by analyzing Reddit posts. While public interest in the use of AI applications in relation to mental health has grown rapidly since ChatGPT’s release, people expressed mixed feelings by discussing both benefits and challenges around these technologies. We also found that perspectives on such advantages and challenges varied significantly based on individual experiences and community contexts. These findings provide valuable data about how people use, perceive, and discuss AI in mental health contexts, offering directions for future research and informing communication strategies for discussions on the benefits and challenges of AI-based mental support across diverse stakeholders.

### 4.1. Growing Attention to AI for Mental Health

The advent of large language models such as ChatGPT marks a pivotal moment in public interest surrounding AI-based mental health support systems. Since ChatGPT’s launch in November 2022, discussions about AI for mental health well-being have surged across both AI and mental health communities. This trend may highlight growing public expectation and interest in advanced AI technologies that can provide mental health support with engaging conversation capabilities.

Large language models (LLMs) have emerged as one of the dominant applications when people discuss AI in the context of mental health. Moreover, a substantial number of users continue to engage with character-based AI platforms such as Character.AI and Replika for mental health support purposes. In contrast, references to therapy bots and other types of chatbots have notably declined since ChatGPT’s release. Growing interests in advanced AIs may reflect the promising capabilities of generic LLMs (e.g., ChatGPT) and LLM-based applications (e.g., Character.AI) in handling users’ queries on support requests ([24]). However, this also raises a safety concern about using such models that are not designed and verified for mental health support, a point raised in multiple studies ([7]; [32]).

The mixed reactions to AI technologies have been captured in our data, demonstrating sentiment reactions toward AI have become more nuanced and complex. The post-ChatGPT landscape has seen more emotionally charged responses to AI technologies in the mental health context, encompassing both enthusiasm and skepticism. This polarized reception indicates that while many users embrace AI’s potential benefits for mental health, a significant portion harbors reservations about its applications in this sensitive domain.

### 4.2. Discourse on AI in Mental Health: Benefits, Challenges, and Research Needs

Our investigation of benefit and challenge discourse provides a deeper understanding of the nuanced and mixed feelings about AI related to mental health. First, users recognized the positive aspects of AI-driven mental health support, particularly commending AI’s ability to provide emotional support and companionship with increased accessibility and availability. Recent advancements in AI’s ability to have natural and contextualized conversations with empathetic responses might facilitate people to recognize and discuss such benefits ([2]; [19]). Recent studies show promising results such that advanced AI chatbots using LLMs significantly reduce loneliness ([10]) and mental disorders such as depression and anxiety ([11]). AI-driven support systems can provide valuable resources for individuals with limited access to support resources, as it can complement human social support.

However, users also highlighted challenges. Adverse effects emerged as a significant concern. Our post hoc analysis (Appendix A) revealed that character-based AI applications were significantly associated with adverse effects (Appendix A) and over-reliance (Appendix A), while other AI services like LLMs and therapy bots showed weaker associations with these problems. This pattern raises specific concerns about the potential risks of character-based AI applications in mental health contexts. Previous research examining social media posts from a Replika community suggests that harms from AI interaction may parallel those from human–human interaction ([16]). While this offers valuable insight, further research is needed to understand the distinct dynamics of AI–human relationships and their long-term implications for mental health. AI–human relationships likely present unique challenges due to the subordinate nature of AI systems (e.g., programmed to avoid contradicting users), their lack of subjective experience (e.g., simulating rather than experiencing empathy), and their commercial design incentives (e.g., optimized for engagement rather than wellbeing). We encourage future research to investigate how these distinct features may shape mental health outcomes over time.

### 4.3. Discourse Variations and Their Implications for Strategic Communication on AI in Mental Health

The pattern of discussing benefits and challenges of AI systems varied by user experience and community type. Experienced users compared to non-experienced users more frequently acknowledged both benefits and limitations of these technologies. They valued AI’s capacity to facilitate self-reflection, provide companionship, and deliver emotional support, while also recognizing its potential to trigger adverse effects. Discussion patterns varied between the AI and mental health communities as well. The AI community highlighted positive aspects more frequently, whereas the mental health community emphasized AI’s limitations in having in-depth conversations and forming emotional connections.

These discrepancies imply a need for tailored communication about the benefits and challenges of AI use for mental health. AI novices often lack comprehensive understanding of how these technologies might help or harm them ([20]). The fact that non-users are less likely than experienced users to discuss both benefits and limitations may indicate that they pay little attention to these features ([4]), potentially due to limited experience and understanding. This highlights the importance of setting clear expectations and guidelines about AI’s potential impacts on psychological well-being for newcomers.

Communication needs to be tailored to specific community contexts as well. Our data show the AI community compared to the mental health community tend to emphasize positive aspects of AI technologies. Members in the AI community, being tech savvy, may overlook potential psychological risks due to their general positive attitudes toward technologies ([14]). Targeted communications need to help them recognize and mitigate negative psychological impacts.

Communicating about AI for mental health requires exceptional caution within mental health communities. Individuals with mental health conditions may be particularly vulnerable to AI interactions, both due to psychological vulnerabilities and social isolations. The risk of developing emotional dependence on the AI system can magnify the harm of misleading information, especially among those who may experience periods of difficulty distinguishing between constructive and unconstructive responses ([8]). The potential for AI to inadvertently worsen mental health, reinforce distorted thinking, or foster unhealthy reliance underscores the critical importance of transparent communication about these risks with this population ([10]).

While acknowledging limitations is essential, clearly communicating the potential benefits of AI is also important. Recent clinical research shows that advanced AI systems, such as LLMs, when reliably designed for therapeutic purposes, can effectively address certain mental health conditions, including depression, anxiety, and eating disorders ([11]). Our data indicate that members of mental health communities tend to rely on more conventional AI systems such as therapy chatbots and others compared to those in AI-focused communities (Appendix A). This reliance on older technologies may lead users to share experiences based on outdated AI capabilities, creating inaccurate impressions among community members. Consequently, such misrepresentations may foster unnecessary hesitancy toward potentially beneficial AI tools, ultimately limiting access to valuable technological support for those who might benefit most. Interdisciplinary collaboration across academia, government institutions, and the IT and health industries is required to develop AI mental health systems that are safe, reliable, and trustworthy.

### 4.4. Limitations

While using social media data enables the collection of unobtrusive responses and helps address the challenges of recruiting participants who have interacted with AI for mental health support over time, this method has inherent limitations. First, the findings should not be generalized to the broader population. This study collected data from a specific social media platform and focused on particular communities. Moreover, the sample may be biased toward individuals who are more tech-savvy and comfortable sharing their experiences online. This lack of representativeness and potential sampling bias limits the generalizability of our findings to populations that do not use Reddit, are less engaged with technology, or are less likely to express their opinions publicly. Future research should explore the benefits and challenges of AI use in mental health with more representative samples.

Second, our sole reliance on social media data limits our ability to understand how individual characteristics shape experiences with AI in mental health contexts. The way people experience and discuss their interactions with AI may vary based on individual factors such as demographics, attachment tendency, and perceptions of AI ([10]; [28]). However, our dataset lacks this kind of contextual information, leaving important questions unanswered about who discusses these experiences and how they frame them. Future research should incorporate more comprehensive data sources—such as surveys or interviews—that capture individual-level variables to better contextualize and interpret online discourse ([18]).

### 4.5. Conclusions

This research provides valuable insights into the evolving landscape of AI applications for mental health support. Our findings demonstrate that since ChatGPT’s release, public interest in AI mental health applications has increased substantially, with a notable shift toward more advanced AI systems, including LLMs and character-based platforms. Users predominantly value AI for its ability to provide emotional support, companionship, and accessibility, though concerns about adverse effects and insufficient emotional depth remain significant. The variation in perspectives between experienced and inexperienced users, as well as between AI and mental health communities, underscores the need for tailored communication strategies that address each group’s specific knowledge gaps and concerns. As AI continues to advance, it will be essential to set clear expectations for novice users and highlight potential risks for technology enthusiasts. While special caution is required when engaging mental health communities, equitable access to information about emerging capabilities must also be ensured so that individuals can make informed decisions about whether and how to benefit from such systems.

Most importantly, safety must remain the highest priority. Given that mental health exists on a continuum and AI’s behaviors can be unpredictable—potentially affecting anyone regardless of their baseline psychological state—comprehensive guidance warning of potential negative impacts should be provided to all users until we have rigorous evidence of AI safety in mental health contexts. This guidance should include clear information about when to seek professional help, recognition of AI limitations, and strategies for healthy engagement with AI systems. We recommend that AI developers rigorously evaluate their systems, transparently communicate the challenges associated with AI use in mental health, and collaborate with clinicians and policymakers to ensure responsible implementation. Future research should prioritize the development of standardized protocols for safe, reliable, and human-centered AI support systems that complement traditional care while minimizing potential harms across all user populations.

## Figures and Tables

**Figure 1 behavsci-15-01172-f001:**
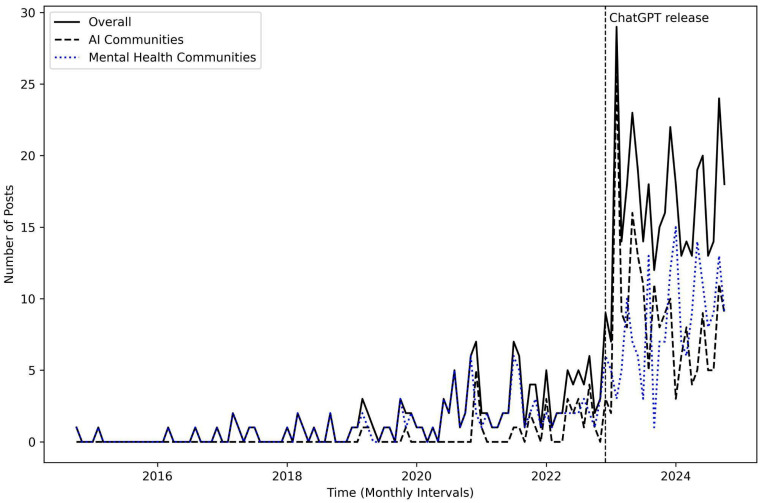
Trend of posts discussing AI and mental health.

**Figure 2 behavsci-15-01172-f002:**
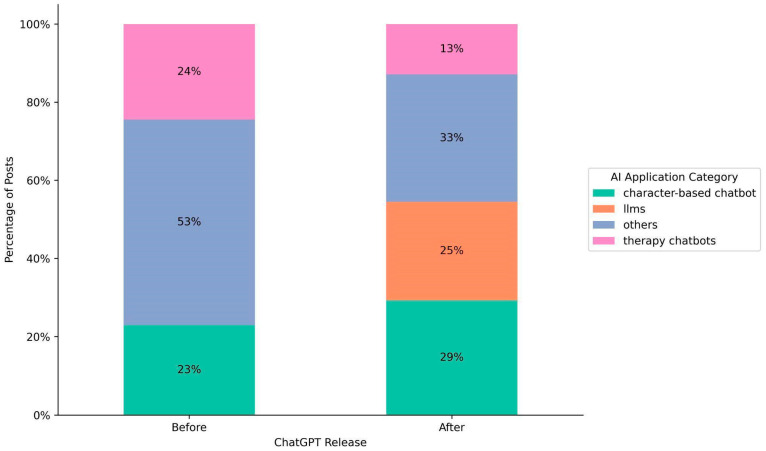
Distribution of AI applications mentioned in posts before and after ChatGPT release. Before ChatGPT’s release, most posts discussed “other” AI applications (53%), followed by therapy chatbots (24%) and character-based chatbots (23%). After the release, posts became more diversified, with notable growth in mentions of large language models (25%) and character-based chatbots (29%), alongside a decline in “other” AI applications (33%) and therapy chatbots (13%).

**Figure 3 behavsci-15-01172-f003:**
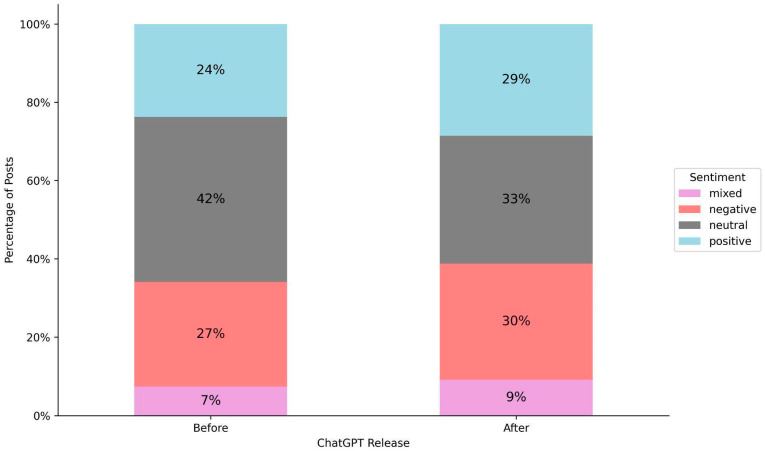
Distribution of sentiment toward AI expressed in posts before and after ChatGPT release. While neutral sentiment remained the most common in both periods, it dropped from 42% to 33% after ChatGPT’s release. Positive sentiment increased from 24% to 29%, while negative sentiment rose slightly from 27% to 30%, and mixed sentiment also saw a small increase from 7% to 9%.

**Figure 4 behavsci-15-01172-f004:**
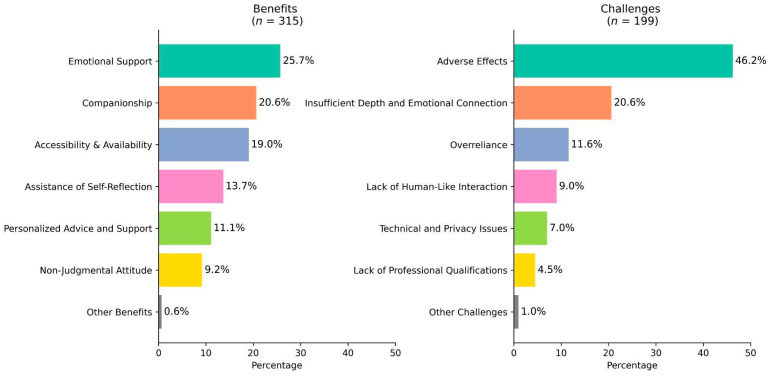
Benefits and challenges of AI use regarding mental health.

**Table 1 behavsci-15-01172-t001:** Potential benefits and limitations of AI in mental health.

Benefits and Challenges	Category	Definitions	Source
Benefits	Accessibility and Availability	The ease of access to AI support and its round-the-clock availability.	[23] ([23]); [26] ([26]); [31] ([31])
	Assisting Self-Reflection	AI’s ability to encourage users to reflect on their own thoughts, emotions, and behaviors.	[23] ([23])
	Companionship	AI’s ability to provide a sense of presence or companionship, helping individuals feel less lonely.	[23] ([23])
	Emotional Support	AI’s capacity to provide comfort, empathy, and encouragement, helping users feel understood and supported.	[12] ([12]); [17] ([17])
	Non-judgmental Attitude	AI’s neutrality and lack of judgment, making users feel comfortable expressing themselves openly without fear of being judged.	[17] ([17]); [20] ([20]); [23] ([23]); [26] ([26]); [31] ([31])
	Personalized Advice and Support	AI’s capacity to provide tailored responses based on individual needs or preferences.	[12] ([12]); [31] ([31])
	Other Benefits	Benefits that fall outside the defined category.	-
Challenges	Adverse effects	Negative outcomes caused by AI interactions, such as psychological harm, relational distortions, or emotional destabilization.	[17] ([17]); [23] ([23])
	Insufficient Depth and Emotional Connection	AI’s failure to engage in emotionally resonant, contextually aware dialogue.	[17] ([17]); [23] ([23]); [26] ([26]); [31] ([31])
	Lack of Human-like Interaction	AI’s failure to replicate the natural flow, spontaneity, and social richness of human communication.	-
	Lack of Professional Qualifications	AI’s lack of professional expertise or qualifications, leading to inappropriate or unhelpful responses, particularly in the context of mental health.	-
	Over-reliance	Behavioral dependence on AI as a substitute for human agency or connection.	[12] ([12]); [23] ([23])
	Technical and Privacy Issues	Technical issues and privacy concerns related to AI in mental health.	[17] ([17]); [20] ([20]); [31] ([31])
	Other Challenges	Challenges that fall outside the defined category.	-

**Table 2 behavsci-15-01172-t002:** Definitions and classes of variables.

Variables	Definitions	Classes
Relevance	Refers to posts where users share experiences using language-based AI tools (e.g., chatbots or apps) that affect their emotional or psychological well-being, or discuss the idea of using AI for mental health—even if they have not used the tool personally.	Yes: The post focuses on any interaction with AI tools that might affect emotional states, highlights specific mental health impacts (e.g., reducing loneliness, causing distress), or explicitly identifies AI tools designed for mental health support. No: The post discusses AI without any connection to mental or emotional health. Unknown: There is insufficient information to determine whether the discussion of AI relates to mental or emotional well-being (e.g., the post is too brief or the connection is ambiguous).
Promotion and prompt	Refers to posts that primarily promote AI applications or services, encourage users to try them, or share AI-generated prompts without adding personal opinions or critical discussion.	Yes: The post explicitly promotes AI applications or services, asks users to try them, or shares AI-generated prompts without personal evaluation. No: The post does not focus on promotion or prompting AI use.
AI applications	Refers to language-based AI applications mentioned in posts.	Character-based chatbots: Examples include applications like character.ai and Replika. LLMs: Large Language Model–based applications such as ChatGPT, Gemini, Claude, and Llama. Therapy chatbots: AI applications specifically designed for mental healthcare (e.g., Wysa, Woebot, Youper). Others: Language-based AI applications that are unspecified (e.g., AI bots) or do not fit into the above categories.
User interaction	Refers to instances where the author directly interacts with AI tools that affect their mental or emotional well-being.	Yes: The author describes using or having used AI-based tools (e.g., chatbots, AI companions) and acknowledges some impact on their own mental or emotional state. No: The author discusses only the concept or general impact of AI on mental health, or references others’ experiences, without mentioning personal interaction. Unknown: It is unclear whether the author directly uses or interacts with such AI tools.
Dominant sentiment toward AI	Refers to the author’s predominant sentiment regarding AI applications in the context of mental health.	Positive: Predominantly favorable or beneficial views toward AI. Negative: Predominantly unfavorable or harmful views toward AI. Mixed: Both positive and negative views are expressed to a similar degree. Neutral: Observations are stated without a clear positive or negative sentiment toward AI.

**Table 3 behavsci-15-01172-t003:** Association between AI experience and mentions of benefits and challenges.

Variables		*b* [95% CI]	Odds Ratio [95% CI]	*p*-Value
Benefits	Assisting Self-reflection	1.93 [0.35, 4.82]	6.87 [1.42, 123.77]	0.06
	Accessibility and Availability	0.56 [−0.35, −1.66]	1.74 [0.70, 5.28]	0.27
	Companionship	2.40 * [0.83, 5.28]	10.98 [2.3, 197.16]	0.02
	Emotional Support	1.18 * [0.22, 2.42]	3.26 [1.24, 11.21]	0.03
	Non-judgmental	1.28 [−0.33, 4.18]	3.59 [0.72, 65.17]	0.22
	Personalized Advice	0.90 [−0.35, 2.75]	2.47 [0.70, 15.65]	0.23
	Other Benefits	- [-]	- [-]	0.997
Challenges	Adverse Effects	0.95 [0.05, 2.05]	2.58 [1.05, 7.77]	0.06
	Insufficient Depth and Emotional Connection	1.87 [0.29, 4.76]	6.46 [1.33, 116.39]	0.07
	Lack of Human-like Interaction	- [-]	- [-]	0.99
	Lack of Professional Qualifications	- [-]	- [-]	0.99
	Overreliance	- [-]	- [-]	0.99
	Technical and Privacy Issues	- [-]	- [-]	0.99
	Other Challenges	−1.90 [−5.15, 1.34]	0.15 [0.01, 3.83]	0.18

Note: * *p* < 0.05. Posts from users without AI experience in mental health served as the reference group. Positive coefficients indicate that posts from users with AI experience are more likely to include the corresponding aspects compared to those from users without such experience. Statistics—including coefficients, odds ratios, and 95% confidence intervals—are not reported for variables affected by quasi-complete separation, leading to unstable and inflated estimates.

**Table 4 behavsci-15-01172-t004:** Association between community type and mentions of benefits and challenges.

Variables		*b* [95% CI]	Odds Ratio [95% CI]	*p*-Value
Benefits	Assisting Self-reflection	−1.13 ** [−1.89, −0.44]	0.32 [0.15, 0.64]	0.002
	Accessibility and Availability	−0.76 * [−1.35, −0.19]	0.47 [0.26, 0.83]	0.01
	Companionship	−1.17 * [−1.79, −0.59]	0.31 [0.17, 0.55]	0.0001
	Emotional Support	−0.95 *** [−1.48, −0.43]	0.39 [0.23, 0.65]	0.0004
	Non-judgmental	−0.98 * [−1.88, −0.18]	0.37 [0.15, 0.84]	0.02
	Personalized Advice	−1.10 ** [−1.94, −0.35]	0.33 [0.14, 0.71]	0.006
	Other Benefits	- [-]	- [-]	0.996
Challenges	Adverse Effects	−0.45 [−0.93, 0.02]	0.64 [0.39, 1.02]	0.06
	Insufficient Depth and Emotional Connection	0.91 ** [0.24, 1.64]	2.49 [1.27, 5.13]	0.009
	Lack of Human-like Interaction	0.29 [−0.66, 1.27]	1.34 [0.52, 3.57]	0.55
	Lack of Professional Qualifications	- [-]	- [-]	0.993
	Overreliance	−0.61 [−1.54, 0.25]	0.54 [0.21, 1.28]	0.17
	Technical and Privacy Issues	- [-]	- [-]	0.996
	Other Challenges	- [-]	- [-]	0.996

Note: * *p* < 0.05, ** *p* < 0.01, *** *p* < 0.001. Posts from AI community members served as a reference. Positive coefficients indicate that posts from mental health community members are more likely to include the corresponding aspects compared to posts from AI community members. Statistics—including coefficients, odds ratios, and 95% confidence intervals—are not reported for variables affected by quasi-complete separation, leading to unstable and inflated estimates.

## Data Availability

We share a portion of the Reddit data along with our content analysis results. Researchers can locate original using submission IDs. The data and code are shared on our Open Science Framework (OSF) project page at https://osf.io/nfx7b/ (accessed on 21 June 2025).

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
