# Peer review of "The ChatGPT Effect: Investigating Shifting Discourse Patterns, Sentiment, and Benefit–Challenge Framing in AI Mental Health Support"

_behavsci, 2025, doi:10.3390/bs15091172_

Round 1
Reviewer 1 Report
Comments and Suggestions for Authors
This is a very interesting work that I have been personally considering quite a lot lately. The introduction is well written and related with past research with a good flow. I find the table you introduce in the introduction slightly unusual as there is no record how this structured information was precisely synthetized. What remans confusing for me is how model coding reliability was validated in the first place? Another thing I would advise as mh expert is elaboration on ethical risks on "emotional dependence" of the users, particularly those at higher risk from inaccurate information (e.g., individuals living with psychosis or schizophrenia would experience periods where they would find hard to distinguish accuracy of the chatgpt responses). Consider explaining "non-judgemental" 0.78 outlier in the main text (most values indicate moderate effects except this one) I would advise to mention early on strong cramer's v values. Slight deviation from the main body but interesting point to consider, is everyone in a proper mental health state to independently decide they can actually receive sufficient help from chatgpt for instance or should there be some guidance on that as well? The constructs in Table 4 would benefit from better clarification in terms of bias or rather validity of such codes, was validation by the authors used after these codes/categories were developed to ensure inter-rater validation?
Author Response
Responses to Reviewer 1 (R1)
R1-1: This is a very interesting work that I have been personally considering quite a lot lately. The introduction is well written and related with past research with a good flow.
Response to R1-1: We sincerely appreciate the reviewer’s positive and constructive feedback on the manuscript. Below, we provide detailed, point-by-point responses to the reviewer’s comments.
R1-2: I find the table you introduce in the introduction slightly unusual as there is no record how this structured information was precisely synthetized.
Response to R1-2: We thank the reviewer for this constructive feedback. We agree that presenting Table 1 without information about how we derived the items was confusing. We identified most items through a narrative review while some items emerged through our data analysis. To make this clear, we added this information as shown below.
Added sentences (Page 3): Table 1 lists the benefits and limitations along with their definitions. Most items (n = 10) were identified through a narrative review of literature discussing benefits and challenges of AI use in mental health, while others were derived from our own data analysis (n = 4; detailed descriptions are available in the methods section). Below, we provide narrative descriptions of how each category benefits or poses risks for individuals who use AI for mental healthcare.
R1-3: What remans confusing for me is how model coding reliability was validated in the first place?
Response to R1-3: We acknowledge that the current version is insufficient to establish the reliability of the model’s coding. To address this concern, we conducted an additional validation by training a fine-tuned GPT-4o model on a subset of the ground truth data (n = 50) and evaluating it on the remaining held-out data (n = 20). Because the test data was never used in training, this evaluation provides an unbiased assessment of coding accuracy.
The subset-trained model achieved an average accuracy of 0.87, with precision, recall, and F1-scores of 0.71, 0.71, and 0.70, respectively. These metrics indicate reliable performance of the fine-tuned model on the coding task. Because model performance typically improves with larger training datasets, we infer that the model used in the main analysis, which was trained on the complete ground truth dataset (n = 70), is likely to perform at least as well, if not better. We have added this validation result to the manuscript and provided the full results in Table S4. We thank the reviewer for this constructive comment, which has helped us strengthen the methodological rigor of our study.
The revised paragraphs (page. 6): “The human-annotated data served as ground truth to fine-tune GPT-4o. To assess model performance, we trained a GPT-4o model on a subset of the ground truth data (n = 50), reserving the remaining samples (n = 20) as a true holdout test set. On this holdout, the model achieved an accuracy of 0.87, with precision, recall, and F1-scores of 0.71, 0.71, and 0.70, suggesting the model could generalize to unseen data (See Table S4 for details). To make full use of the available annotations, we then fine-tuned GPT-4o on the complete ground truth dataset (n = 70) and applied this model to code the full dataset. This model achieved a training loss of 0.001 and validation loss of 0.032. While interpretation of these values is task-dependent, the relatively low and similar losses may suggest that the model fit the data effectively without obvious signs of overfitting”
R1-4: Another thing I would advise as mh expert is elaboration on ethical risks on "emotional dependence" of the users, particularly those at higher risk from inaccurate information (e.g., individuals living with psychosis or schizophrenia would experience periods where they would find hard to distinguish accuracy of the chatgpt responses).
Response to R1-4: We appreciate the reviewer’s suggestion expanding our discussion on ethical risks on emotional dependence of the users with psychosis or schizophrenia. We absolutely agree that such advisory comment is truly important to warn those who are mentally vulnerable to AI systems. In revision, we expanded our discussion like below
Revised paragraph (page 15): Communicating about AI for mental health requires exceptional caution within mental health communities. Individuals with mental health conditions may be particu-larly vulnerable to AI interactions, both due to psychological vulnerabilities and the risk of developing emotional dependence on the system. Such dependence can magnify the harm of inaccurate or misleading information—especially for individuals with psychosis or schizophrenia—who may experience periods of difficulty distinguishing between accurate and inaccurate responses from AI systems (Dohnány et al., 2025). The potential for AI to inadvertently worsen mental health, reinforce distorted thinking, or foster un-healthy reliance underscores the critical importance of transparent communication about these risks when engaging this population (Fang et al., 2025).
Added reference:
Dohnány, S., Kurth-Nelson, Z., Spens, E., Luettgau, L., Reid, A., Summerfield, C., ... & Nour, M. M. (2025). Technological folie\a deux: Feedback Loops Between AI Chatbots and Mental Illness. arXiv preprint arXiv:2507.19218.
R1-5: Consider explaining "non-judgemental" 0.78 outlier in the main text (most values indicate moderate effects except this one) I would advise to mention early on strong cramer's v values.
Response to R1-5: We appreciate the reviewer’s careful attention to the information in the supplemental material. Upon review, we identified that this was a typographical error rather than an outlier. The correct Cramer’s V value is 0.28, not 0.78. We have corrected this in the revised version.
R1-6: Slight deviation from the main body but interesting point to consider, is everyone in a proper mental health state to independently decide they can actually receive sufficient help from chatgpt for instance or should there be some guidance on that as well?
Response to R1-6: We appreciate the reviewer's thoughtful and interesting comment. We believe that individuals with proper mental health still need guidance on AI use in mental health contexts for two important reasons.
First, defining what constitutes "proper mental health" is inherently complex and multifaceted. Mental health exists on a continuum, and individuals who appear mentally well may still have underlying vulnerabilities that are not immediately apparent. For example, someone may not currently experience symptoms of depression but could still be at risk during stressful periods or life transitions. Additionally, people may have different coping mechanisms and resilience levels that affect how they respond to AI interactions. Given this complexity, it is difficult to predict how AI-based support systems will affect individuals across varying mental health states and circumstances.
Second, even individuals with robust mental health can experience harm from AI systems. According to the Computers Are Social Actors (CASA) paradigm, humans naturally treat technologies as social entities, making them susceptible to psychological impacts similar to those from human interactions. AI systems can exhibit unpredictable behaviors, provide inappropriate advice, or create problematic usage patterns that can negatively affect anyone, regardless of their baseline mental health status. For instance, AI might reinforce unhelpful thinking patterns, provide conflicting information that creates confusion, or inadvertently encourage over-reliance on technology rather than human support when needed.
Given these considerations and the current early stage of AI development in mental health contexts, we believe that guidance informing users of AI limitations and potential risks should be provided to anyone considering its use, including those with proper mental health.
We have integrated this discussion into our conclusion to strengthen the manuscript:
Revised conclusion (page 16): “…Most importantly, safety must remain the highest priority. Given that mental health exists on a continuum and AI's behaviors can be unpredictable—potentially affecting anyone regardless of their baseline psychological state—comprehensive guidance warning of potential negative impacts should be provided to all users until we have rigorous evidence of AI safety in mental health contexts. This guidance should include clear information about when to seek professional help, recognition of AI limitations, and strategies for healthy engagement with AI systems. We recommend that AI developers rigorously evaluate their systems, transparently communicate the challenges associated with AI use in mental health, and collaborate with clinicians and policymakers to ensure responsible implementation. Future research should prioritize the development of standardized protocols for safe, reliable, and human-centered AI support systems that complement traditional care while minimizing potential harms across all user popula-tions.”
R1-7: The constructs in Table 4 would benefit from better clarification in terms of bias or rather validity of such codes, was validation by the authors used after these codes/categories were developed to ensure inter-rater validation?
Response to R1-7: We appreciate the reviewer's thoughtful comment regarding the validity and potential bias of our coding tasks. We acknowledge that our original description of the coding procedure was insufficient and lacked clarity about validation measures.
While we did not calculate traditional inter-rater reliability statistics (e.g., Cohen's kappa), we employed a rigorous consensus-based validation approach specifically designed for ground truth dataset development. Three independent coders initially coded a subset of posts. Through discussion of discrepancies, they iteratively refined the coding scheme until achieving complete consensus on coding criteria and application. This process served as a form of content validation, ensuring that our coding scheme accurately captured the intended constructs and could be consistently applied. Once consensus was established, all remaining posts were coded jointly using the finalized scheme, with any disagreements resolved through immediate discussion. This collaborative and consensus-based approach ensured that potential biases were minimized and that the coding scheme was consistently applied across the dataset. We have expanded the methodology section to clarify this coding process.
Added sentences (page 6): “…Creating a fine-tuned model requires training data that adequately represents the var-iables of interest. To generate such data, we first used a generic GPT-4o model to identify posts potentially relevant to our study (i.e., discussing AI in mental health contexts). From these, we sampled a subset of Reddit posts (n = 70) for manual coding. Three coders independently coded an initial subset of these posts. Discrepancies were discussed, and the coding scheme was refined until consensus was achieved. The remaining posts were then jointly coded using the finalized scheme, with disagreements resolved through discussion.”
Reviewer 2 Report
Comments and Suggestions for Authors
This is an outstanding and timely manuscript that investigates the evolving public discourse on AI-based mental health support, particularly before and after the release of ChatGPT. The authors demonstrate a well-designed methodology, including a robust mixed-methods approach that leverages naturalistic Reddit data and a fine-tuned GPT-4o model for content analysis.
Strengths:
-
The introduction provides a thorough overview of the literature and clearly establishes the study’s relevance.
-
Research questions are well-defined and logically flow from the background.
-
The use of Reddit posts and the stratification by AI experience and community type provides rich, novel insights.
-
The figures and tables (especially Table 1 and Figures 1–4) are informative and support the claims well.
-
The discussion demonstrates thoughtful interpretation, especially regarding implications for tailored communication strategies and ethical AI integration.
Suggestions:
-
Methodology Clarification: While the GPT-4o fine-tuning and coding are well explained, a bit more clarity on how posts were selected as valid (e.g., what qualified a post as “emotionally impactful” or “mental health-relevant”) could improve reproducibility.
-
Ethical Considerations: A brief elaboration on how ethical considerations were addressed when using publicly available Reddit data (even if exempt from IRB) would strengthen the rigor.
-
Figure Captioning: Some figures (e.g., Figure 2 and 3) would benefit from more descriptive captions that interpret the trends rather than just describe the content.
-
Conclusion Clarity: The conclusion is strong but could benefit from a short summary of policy or practical recommendations (e.g., “We recommend…” or “Developers should…”).
Overall, this work significantly contributes to the understanding of public attitudes toward AI in mental health and highlights essential future research and implementation considerations.
Author Response
Responses to Reviewer 2 (R2)
R2-1: This is an outstanding and timely manuscript that investigates the evolving public discourse on AI-based mental health support, particularly before and after the release of ChatGPT. The authors demonstrate a well-designed methodology, including a robust mixed-methods approach that leverages naturalistic Reddit data and a fine-tuned GPT-4o model for content analysis.
Strengths:
The introduction provides a thorough overview of the literature and clearly establishes the study’s relevance.
Research questions are well-defined and logically flow from the background.
The use of Reddit posts and the stratification by AI experience and community type provides rich, novel insights.
The figures and tables (especially Table 1 and Figures 1–4) are informative and support the claims well.
The discussion demonstrates thoughtful interpretation, especially regarding implications for tailored communication strategies and ethical AI integration.
Response to R2-1: We appreciate the reviewer’s positive feedback on the manuscript as well as thoughtful and constructive suggestions. Below, we address the reviewer’s concerns point by point.
R2-2: Suggestions: Methodology Clarification: While the GPT-4o fine-tuning and coding are well explained, a bit more clarity on how posts were selected as valid (e.g., what qualified a post as “emotionally impactful” or “mental health-relevant”) could improve reproducibility.
Response to R2-2: We agree that our explanation is insufficient for readers to understand how coders evaluate content and construct a ground truth dataset used for fine-tuning GPT-4o. As it is important that a ground truth dataset includes sufficient presence of the variables of interest, we first used a generic GPT-4o to roughly identify relevant posts that discuss language-based AI systems in the context of mental health. Then, we subset data (n = 70) among posts that a generic GPT-4o identified as relevant, three coders reviewed, evaluated, and discussed each post to code variables based on definitions and class criteria presented in Table 2. We did not elaborate this procedure in the initial manuscript. In the revision, we included such information. Additionally, in response to another reviewer’s concern regarding the validity of the fine-tuned GPT-4o model, we performed a validity test, which is now described in the methods section. We thank both reviewers for their helpful suggestions, which improved the methodological rigor of the research.
Added sentences (page. 6): “Creating a fine-tuned model requires training data that adequately represent the vari-ables of interest. To generate such data, we first used a generic GPT-4o model to identify posts potentially relevant to our study (i.e., discussing AI in mental health contexts). From these, we sampled a subset of Reddit posts (n = 70) for manual coding. Three coders independently coded an initial subset of these posts. Discrepancies were discussed, and the coding scheme was refined until consensus was achieved. The remaining posts were then jointly coded using the finalized scheme in Table 2, with disagreements resolved through discussion.
The human-annotated data served as ground truth to fine-tune GPT-4o. To assess model performance, we trained a GPT-4o model on a subset of the ground truth data (n = 50), reserving the remaining samples (n = 20) as a true holdout test set. On this holdout, the model achieved an accuracy of 0.87, with precision, recall, and F1-scores of 0.71, 0.71, and 0.70, suggesting the model could generalize to unseen data (See Table S4 for details). To make full use of the available annotations, we then fine-tuned GPT-4o on the complete ground truth dataset (n = 70) and applied this model to code the full dataset. This model achieved a training loss of 0.001 and validation loss of 0.032. While interpretation of these values is task-dependent, the relatively low and similar losses may suggest that the model fit the data effectively without obvious signs of overfitting.”
R2-3: Ethical Considerations: A brief elaboration on how ethical considerations were addressed when using publicly available Reddit data (even if exempt from IRB) would strengthen the rigor.
Response to R2-3: We appreciate the reviewer’s thoughtful comment. We fully agree with the suggestion to include ethical considerations regarding the collection and processing of data. In response, we added a paragraph in the Methods section elaborating on how we collect, store, and handle data to minimize potential risks of individual identification. The added paragraph reads as follows:
Added paragraph (page 5): “IRB review was exempted for collecting publicly accessible Reddit posts in which all content is viewable without registration or login. No interaction with users occurred, and no private or direct messages were accessed. All data is stored on a local computer pro-tected by password authentication, with usernames removed before data analysis. We did not report any individual posts in our findings to prevent potential identification of users. All results are presented as aggregated data only, ensuring individual voices cannot be traced back to specific users.”
R2-4: Figure Captioning: Some figures (e.g., Figure 2 and 3) would benefit from more descriptive captions that interpret the trends rather than just describe the content.
Response to R2-4: In line with the reviewer’s suggestion, we have added statements that describe the trends shown in Figures 2 and 3. The added statements are as follows.
Revision of Figure 2 (Page 9): Figure 2. Distribution of AI Applications Mentioned in Posts Before and After ChatGPT Release. Before ChatGPT’s release, most posts discussed “other” AI applications (53%), followed by thera-py chatbots (24%) and character-based chatbots (23%). After the release, posts became more di-versified, with notable growth in mentions of large language models (25%) and character-based chatbots (29%), alongside a decline in “other” AI applications (33%) and therapy chatbots (13%).
Revision of Figure 3 (Page 10): Figure 3. Distribution of Sentiment toward AI Expressed in Posts Before and After ChatGPT Re-lease. While neutral sentiment remained the most common in both periods, it dropped from 42% to 33% after ChatGPT’s release. Positive sentiment increased from 24% to 29%, while negative sen-timent rose slightly from 27% to 30%, and mixed sentiment also saw a small increase from 7% to 9%.
R2-5: Conclusion Clarity: The conclusion is strong but could benefit from a short summary of policy or practical recommendations (e.g., “We recommend…” or “Developers should…”). Overall, this work significantly contributes to the understanding of public attitudes toward AI in mental health and highlights essential future research and implementation considerations.
Response to R2-5: In response to the reviewer’s suggestion, we added a recommendation statement directed at AI developers. We sincerely appreciate the reviewer’s time and constructive feedback.
Revised conclusion (page 16): “…Most importantly, safety must remain the highest priority. Given that mental health exists on a continuum and AI's behaviors can be unpredictable—potentially affecting anyone regardless of their baseline psychological state—comprehensive guidance warning of potential negative impacts should be provided to all users until we have rigorous evidence of AI safety in mental health contexts. This guidance should include clear information about when to seek professional help, recognition of AI limitations, and strategies for healthy engagement with AI systems. We recommend that AI developers rigorously evaluate their systems, transparently communicate the challenges associated with AI use in mental health, and collaborate with clinicians and policymakers to ensure responsible implementation...”